# The Use of Activities and Resources in Archaeological Museums for the Teaching of History in Formal Education

Ainoa Escribano-Miralles *, Francisca-José Serrano-Pastor and Pedro Miralles-Martínez

Espinardo Campus, Education School, University of Murcia, 30100 Murcia, Spain; fjserran@um.es (F.-J.S.-P.); pedromir@um.es (P.M.-M.)
* Correspondence: ainoa.escribano@um.es

**Abstract:** The research objectives of this paper are to compare the activities which have been prepared in the design of field trips from the perspective of teachers and museum educators, as well as to describe the use of resources and materials from the point of view of educational agents. The research method is quantitative, based on the study of a descriptive comparative cross-sectional survey. The participants are 442 teachers of early years, primary and secondary education, visiting two archaeological museums with their class groups in order to carry out an activity relating to the subject of history. The data collection tool was the MUSELA© questionnaire. The main results show that 60% of the teachers state that they prepare some kind of activities and 70% use some resources within the design of a field trip to an archaeological museum. On the other hand, 94.4% of the museum educators carry out activities using resources in the museum visit. The main conclusion is that the activities which are most used by teachers and educators in the museum (experimentation and artistic workshops, audio-visual observation and viewing tasks and debates or sharing) and by teachers in the classroom space (audio-visual viewing) do not guarantee research activities, analysis or reflection activities.

**Keywords:** museum; field trips; history education

## 1. Introduction

The implementation of the 2030 Agenda and sustainability for the transformation of our world are aims which education bodies and museums have been aware of over the course of recent years. At the 34th ICOM General Assembly [1] the need was established to incorporate sustainability into the orientational framework in the internal and external practices of museums and in educational programing. Recent studies have echoed the use of museums in line with the Sustainable Development Goals [2]. Other studies are aimed at the creation of participatory financing models for museums [3].

The approach described in the Manifesto on Learning Outside the Classroom [4] assumes that schoolwork based on experience in 'real' situations guarantees learning and the fulfilment of pupils' needs. Furthermore, the use of resources from the surrounding environment in order to generate learning contributes to the development of a Quality Education and a more sustainable use of educational resources [4,5]. In this regard, museums are presented as an enlightening resource for the teaching and learning process of history, capable of being used not only via school visits but also by diversifying the broad educational use which museums can offer to formal education.

### 1.1. The Approach of Classroom Museums and Museums for the Classroom

Within the learning of history in school, the museum is positioned as a didactic resource of the first order. The use of museum materials and resources, such as the handling of objects, the use with historical sources, will allow the development of activities that mobilize the development of capacities for learning history. The archaeology museum

space by itself is already a space that arouses curiosity about learning about the passage of time. This approach is understood as the interaction brought about when museums are included as educational resources in the teaching and learning process and museum visits are completed as a systematized activity within the teaching program. This interaction constitutes the use of specific methodologies, strategies and teaching techniques and the organization of spaces and times and the resources used. In this way, the museological element becomes a component of the teaching and learning of the social sciences [6,7].

One specific example of the application of this approach is the educational project 'This class is a museum!", carried out in a class of five-year-olds in early years learning, in which all of the teaching and learning process (based on the focal point of the mass media) was developed via a learning project based on research of the work and historical and artistic heritage of the painter Ramón Gaya. The aim of the project was to discover how the grandparents and family members of the pupils communicated in the 1970s and 1980s using as a cross-cutting theme a visit to the Ramón Gaya museum [7,8]. Table 1 shows the methodological specificity of this approach and of the project in question.

**Table 1.** Methodological specificity of the experiment "This class is a museum!".

| The Classroom Museum for the Globalized Teaching of History in Early Years Education | |
|---|---|
| Educational approach | Classroom museum and museum for the classroom |
| Method | Project-based Learning |
| Teaching sequence | Phase 1 (initial): Motivation, queries, exchange of ideas and proposals<br>Phase 2 (development): Obtaining new information (confrontation with new ideas)<br>Phase 3 (synthesis): Structuring (Reflection and conclusions)<br>Phase 4 (final): Using what has been learned |
| Organization of spaces | The classroom museum: work corners<br>Assembly<br>Purple room. Our grandparents house<br>Red room. Exhibition of paintings<br>Green room. The painter's desk<br>Blue room. The study of drawing and painting |
| Resources | Working with primary sources<br>Cultural heritage<br>Visit to the Ramón Gaya Museum |

Source: Authors' own work based on Escribano-Miralles [9].

How do we perceive museums? What is a museum for us? Museums can be considered as drawers or, better, as display cabinets, which, in spite of being full of elements (objects of a material or immaterial nature), is empty as it lacks meaning for pupils if we do not provide them with an effective learning environment. What, then, should be done in order for this display case full of knowledge to awaken the pupils' full potential? Fontal [10] considers museums to be potential repositories of heritage and the link arising between a person, or a group of people, and an asset. As teachers, we must encourage these links between museums and our pupils with the aim of contributing towards their personal development, leading them to value their historical inheritance and to become critical and active citizens with an awareness of culture so that they may understand the past as a way of understanding the present [11].

For these links to be effective, importance should be attributed to heritage education within the teaching and learning process, taking the teaching model of the heritage which is closest to our proposition. In this case, the "Learner-based Model of Heritage Education" [12] (p. 94) could be taken into account. The learner is the active protagonist of the teaching and learning process; the teacher takes the role of guide in this learning and is responsible for pupils building new learning based on their prior knowledge.

Ultimately, in this approach, the teaching has to extend beyond the four walls of the classroom and establish communication and relationships with other educational agents. The main strategies which lead to an optimal learning scenario are the valuing of the surrounding environment as a source of learning and the opening up of dialogue, research and reflection [10,11]. The dichotomy of classroom museums and museums for the classroom necessarily requires the establishment of a good collaborative relationship between the classroom and museums in order to guarantee that the teaching aims and the improvement of the teaching and learning process for history from the initial stages of education can be achieved.

### 1.2. Museums as a Means of Education

Rivero and Feliu and San Martín and Ortega-Sánchez claim that when we visit museums or archaeological sites, a contextualization is necessary in order to understand them [12,13]. Without understanding, the sequence of procedures for heritage education is breaks down:

> *"Understanding our past, knowing how past civilizations lived, discovering what they ate and how they interacted are common interests for people. Indeed, history, and, therefore, archaeology, generates curiosity. But in order to understand the past in all its complexity we need educational tools which bring us closer to it"* [14] (p. 320).

A study analyzed the synergies produced in the practical community composed of a Natural History Museum and the Department of Early Years Education of the University of Crete [15]. Based on the activities carried out by this community of practice, a study was formed on the evaluation of three educational programs of museums which were designed ad hoc.

Among the results, the changes can be highlighted which have occurred in the activities carried out by the children. In the first place, the children's activities were based on the movement and viewing of the different exhibitions and heritage elements. The advancement and improvement of educational programs led to pupils obtaining new actions and increased opportunities for movement during the museum visit.

As far as the use of resources is concerned, mainly during museum visits, they were reduced to the use of the museum facilities and displays. The new programs, designed via the community of practice, introduce different resources used for a wide variety of activities.

### 1.3. International Research Regarding the Use of Resources and Activities Based on Museum Visits

There is a wide range of research carried out on education in museums. Over the course of recent years, countless studies have been made on the research of the effectiveness of school field trips to museums [15–21].

In the field of heritage education, different studies have focused on the analysis of the educational potential presented by the resources that museums offer within the context of school visits. There are notable studies which analyze the educational resources offered by museums [6,22,23]. The study carried out within the research project Evaluación Cualitativa de Programas Educativos en Museos Españoles (Qualitative Evaluation of Educational Programs in Spanish Museums) (ECPEME) presents as one of its objectives the analysis of the educational programs of 14 Spanish museums in order to improve their quality criteria. The analysis of the museums' resources lies within the scope of the evaluation. The analytical variables are their typology, availability, uses and relevance [24]. The types of resources analyzed in the study are classified as traditional and Information and Communication Technology (ICT) resources. The results show that 64.29% of the traditional resources analyzed (worksheets, notebooks, guides and didactic portfolios) are extremely suitable for the visit, whereas 14.29% are classified as not very suitable. As far as the ICT resources are concerned, interactive and audio-visual materials stand out, of which 35.71% are considered totally suitable, 35.71% extremely suitable, 14.29% not very suitable and 7.14% suitable and 7.14% completely unsuitable.

Other studies have attempted to analyze teachers' or trainee teachers' perceptions regarding the use of heritage as an educational resource [25–28]. The main objective of the study by Felices-De la Fuente, Chaparro-Sainz and Rodríguez-Pérez [29] is: "To ascertain which heritage resources are most highly valued by future teachers, for teaching history through heritage" (p. 3). Among the results proposed by the trainee teachers surveyed is that the most valued resources for the teaching of history are local historical heritage (4.61), followed by museums with a score of 4.49. Additionally, worthy of note are the virtual recreations of museums and centres of heritage interest (4.12). The use of mobile applications is the resource valued least by trainee teachers with a score of 3.65.

On an international scale, there are many studies which attempt to analyze the improvements in learning which have been brought about by applying educational programs between museums and schools, using school visits as an educational resource and planning activities before, during and after the school visit to the museum [15–18,20,30]. Uztemur, Dinc and Acun [21] developed action research based on the application of learning activities for the teaching of social studies in historical sites and museums: "The aim of this study is to determine the usefulness of the teaching activities prepared for effective utilization of museums and historical places in the context of grade seven social studies teaching to increase the efficiency of teaching-learning processes" [21] (p. 252). The main results on the value and importance of the activities concerned the subject of social studies include: "proving that doing a lesson out of school is possible", "lasting learning", "learning more", "increase in class participation" and more historical consciousness among the students "comparing the past and present", "learning history", "preserving history artefacts" and "local awareness and information" (p. 263). Some students' suggestions for improving the quality of the activities were to develop "games", "watch videos in class after the trips", "do preparatory homework", "role playing", "creative drama activities"and "drawing pictures about the activities" (p. 263).

The study by Kisiel [31] examines the development of a collaborative project between a school and a science centre (an aquarium). The degree of effectiveness of the application of educational programs based on the development of a community of practice between museums and schools is evaluated. Based on structured interviews, he posed questions relating to the category of analysis: the success of the outreach activities of the visit.

The study carried out by Ampartzaki et al. [15] is based on the evaluation of three educational programs carried out between schools and museums based on a community of practice. Their study analyses "changes in children's activities" (p. 13) and the "use of resources in the three educational programs". This project was carried out with early years pupils and, following the application of the programs, changes were noted in the activities carried out during the visit to the museum:

> *Despite their young age, in the museum tour, children were restricted to moving between dioramas and just looking at the exhibits. In Versions 1 and 2 a variety of other actions enriched the children's visit and increased opportunities for movement. The simple viewing of the museum dioramas was reduced, and the handling of objects was introduced and then increased. Children were invited to look at a variety of resources and work together in groups* [15] (p. 13).

Basically, the main activity was to "look at dioramas" ($n = 7$) and "walk between sections and dioramas in the museum" ($n = 7$) during the museum tour. With the application of the first version of the educational program, the use of these activities diminished, and others were carried out, such as "working collaboratively in groups" ($n = 2$), "handling objects" ($n = 2$), "drawing and/or writing" ($n = 1$), "moving with heavy steps" ($n = 1$), "moving on all fours" ($n = 1$) and "digging". These activities were maintained with the application of the program in its second version and other activities were also designed, such as "looking at books, photographs, videos, etc." ($n = 2$) and "drawing and/or writing" increased ($n = 6$) [15] (p. 15).

As far as the use of resources is concerned, the results of the study determined that the main resource used during the museum tour was the dioramas, "The newly designed

programs introduced other resources that were used for a variety of activities" [15] (p. 20). With the application of programs 1 and 2, other resources were integrated, such as special tools (spades, goggles, brush, clipboard, graph paper and pencil) and sand pits. In addition, in version 2, books, photographs and videos were included.

All of the studies mentioned above are based on qualitative designs. There are no studies which make it possible to carry out a quantitative survey-based design, which attempts to understand the generalized use made by school groups of resources and activities during visits to museums in order to contribute to the learning of history. The work presented here attempts to comprehend which activities and resources are used by teachers for the learning of the social sciences and by educators during visits to archaeological museums.

The study has the main objectives of comparing the activities prepared in the design of field trips from the perspective of teachers and museum educators and describing the use of resources and materials from the point of view of the educational agents.

Specifically, answers are sought to the following research questions: What activities are used by teachers in the museum and in the classroom within the design of the school visit? How often do museum educators use these activities in school visits? What resources and materials do teachers use in the museum and at school within the design of the school visit? How often do museum educators use these resources and materials in school visits?

In the use of these activities, materials and resources proposed by Ampartzaki et al. [15], Rivero and Feliu [32] and Estepa [33] in the Curricular Project Investigando nuestro mundo (6–12), Investigando las sociedades actuales e históricas (Investigating our world. Investigating current and historical societies), the theoretical and empirical study presented has been taken into account, with the aim of establishing the educational possibilities which can be borne in mind within the design of a field trip to an archaeological museum. It is here that the use of resources provided by the museum and activities based on the visit, the object of this study, play a fundamental role.

### 1.3.1. Type of Activities

In the results of King's research, it is pointed out that the typical activities carried out by pupils with the projects that the four museum schools deploy around the school curriculum and the topics or units of the museum are [34]:

> *"observing an object or phenomenon; generating questions, brainstorming and exploration; gathering data, synthesizing findings by creating an exhibit, object, and research papers; disseminating key learnings outside of the classroom through school exhibit openings; presentations, and demonstrations; and reflection" (p. 3).*

Some possible activities are as follows: experimentation or artistic workshops; role plays or theatrical representations; cooperative tasks aimed at problem-solving; treasure hunts; debates; brainstorming; round table discussions; observational tasks and/or the handling of objects; research sequences as fieldwork; data collection activities which imply working with oral history, such as interviewing experts or family members; activities for the presentation of results involving the representation of data; watching audio-visual materials; creating a classroom museum or an exhibition of objects inside the school [4,15].

### 1.3.2. Use of Materials and Resources

The materials and resources which can be used to compliment school field trips to archaeological museums are many and varied. Such materials and resources may include: passive ICT materials (webpages, videos, photographs); active ICT materials (videogames, web applications, QR codes, augmented reality, virtual museums); written and oral sources (books, interviews, people from the surrounding area, etc.); worksheets; artefacts (objects and material remains); models or archaeological reproductions; the use of cartography and maps [35].

All of these activities, materials and resources can be used within the museum space or in the context of the classroom, both in the educational design proposed by the teacher

and in that offered by the museum educator, be it before, during or after the school visit to the archaeological museum.

In the course of their research, Asensio, Santacana and Fontal and Santacana, Martínez Gil, Llonch and López Benito, highlight as one of their objectives the analysis of the perception held by 14–16-year-old pupils regarding the educational activities carried out in museums [16,17]. One of the results demonstrates the lack of interest among adolescents in archaeological museums. The main reason, according to the authors, is the low degree of efficiency which they attribute to the museum resources (text panels, room sheets, illustrations and objects in display cases). In the words of the authors regarding the use of traditional museum resources:

> *"These types of resources do not truly create an atmosphere of dialogue between the museum and its users ( . . . ) the most important thing for a museum to make itself suitable to this type of audience is the predisposition of the museum to interact with them, to transform itself from a passive exhibition into an active laboratory"* [36] (p. 36).

Therefore, in order to encourage pupils to take an interest in and make the most of the museum, it is necessary to take into account the resources used during school trips to the museum, both within the museum space and outside of it in their class work. Transforming the museum itself into a resource, as a learning laboratory, is one of the main challenges towards which advancement should be made.

## 2. Materials and Methods

### 2.1. Defining the Hypotheses

The study analyses the differences between the frequency of use of each of the types of activities carried out by the educators (dependent variable or criterion) and the educational and museological space in which they were carried out (independent variable or predictor) were analyzed.

**Hypothesis 1 (H1).** There are no differences between the frequency of use of each of the types of activities carried out by the educators and the educational museological space in which they are developed.

**Hypothesis 2 (H2).** There are differences between the frequency of use of each of the types of activities carried out by the educators and the educational museological space in which they are developed.

In the same way, the work study differences between the frequency of use of each of the materials and resources employed by the educators (dependent variable or criterion) and the educational museological space in which they are carried out (in-dependent variable or predictor).

**Hypothesis 3 (H3).** There are no statistically significant differences between the frequency of use of each of the materials and resources used by the educators and the educational museological space in which they are carried out.

**Hypothesis 4 (H4).** There are statistically significant differences between the frequency of use of each of the materials and resources used by the educators and the educational museological space in which they are carried out.

### 2.2. Approach and Design

A research design with a quantitative approach has been chosen in order to respond to the problems and objectives posed (Table 2). A cross-cutting survey has been employed with a descriptive-comparative purpose. The study is focused on a descriptive phase in which quantitative information is handled based on questionnaires administered to teachers and museum educators with the aim of attempting to understand how school

visits to archaeological museums are planned. Furthermore, the study of hypothesis testing implies that it is developed at a relational research level.

**Table 2.** Research objectives and analysis variables according to the methodological function in the design of the research.

| Planning for a Field Trip to an Archaeological Museum | | |
|---|---|---|
| Objectives | Independent Variable | Dependent Variable |
| Objective 1. To compare the activities which have been prepared in the design of the field trips from the perspective of teachers and museum educators. | Role of the educational agents The educational space | Type of activities |
| Objective 2. To describe the use of the resources and materials from the point of view of the educational agents. | | Type of resources and materials |

Source: authors' own work.

### 2.3. Context and Participants

This research is set in the context of two archaeological museums which, due to their size, organization, heritage typology, capacity and audience, have common characteristics and constitute an ideal space for the learning of history in the stages of early years education, primary education and compulsory secondary education.

More specifically, the two archaeological museums are located in the Autonomous Community of the Region of Murcia (MU) and the province of Alicante (MA). In spite of the fact that one is provincial and the other is controlled by the Autonomous Community in question, both museums are similar in terms of size, the amount of heritage they accommodate, typology, etc. However, there are also notable differences, MA has a permanent professional team of five staff in the museum, who form the Department of Education and Dissemination.

In order to guarantee anonymity, the exact name of each museum has not been used. Below, we shall present some of the most relevant characteristics of both contexts.

### 2.3.1. The Archaeological Museum Located in Murcia (MU)

The MU is a state-owned institution, the management of which has been transferred to the Autonomous Community of the Region of Murcia. It has been located in the old Palacio Provincial de Archivos, Bibliotecas y Museos since its re-opening in 2007. It has made a great effort to develop its role as an educator and disseminator of heritage.

The permanent collection of the museum proposes a stroll through Pre-history, from the Palaeolithic to the Bronze Age, continuing through Ancient History up to the Visigoth Period. Its discourse is based on three main aspects: types of habitats, the evolution of technology and funeral rites (Figure 1).

### 2.3.2. The Archaeological Museum Located in Alicante (MA)

The MA is a facility whose competences and legal status are structured and managed by two large institutions. The archaeological museum (created in 1932) depends administratively on the Provincial Government of Alicante and the Foundation of the Valencian Community was set up in 2001 at a time of great momentum when the location of the museum was changed, adopting its place in the architectonic complex of the old San Juan de Dios Provincial Hospital, which was inaugurated in 2002.

After occupying its new premises, from 2002, its educational and innovative discourse runs through each of the rooms and installations of the museum. As a result of this, it became necessary to create a permanent team dedicated to dissemination and education in the museum. In 2005, the museum's Department of Education and Dissemination was formed. The competences of this department are outlined below.

The museum consists of five permanent rooms, three thematic rooms, three rooms which are used for temporary exhibitions and the basements. Different archaeological and monumental sites also lie under the management of the MA (Figure 2).

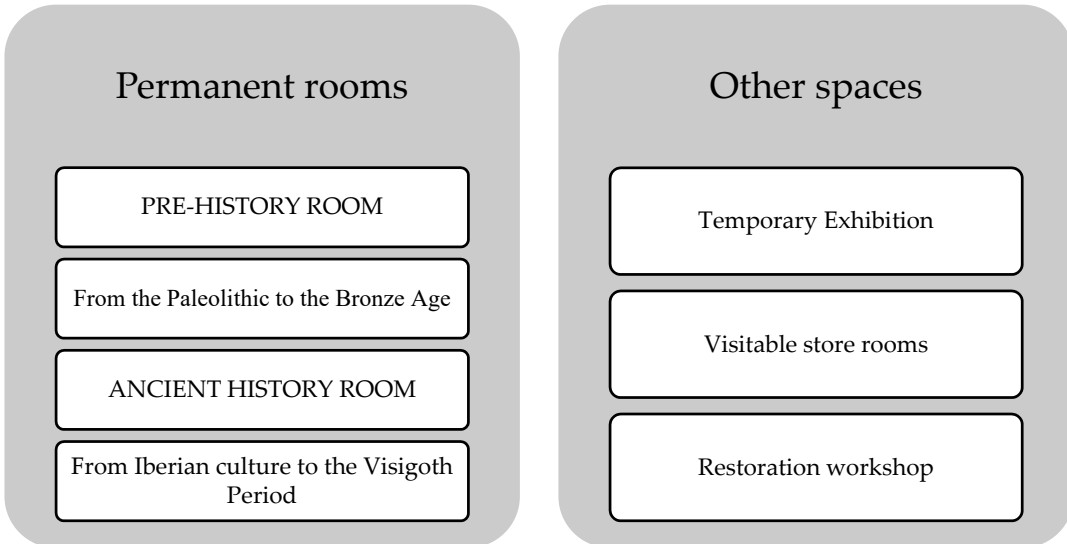

**Figure 1.** Spatial organization of the Murcia museum. Source: authors' own work.

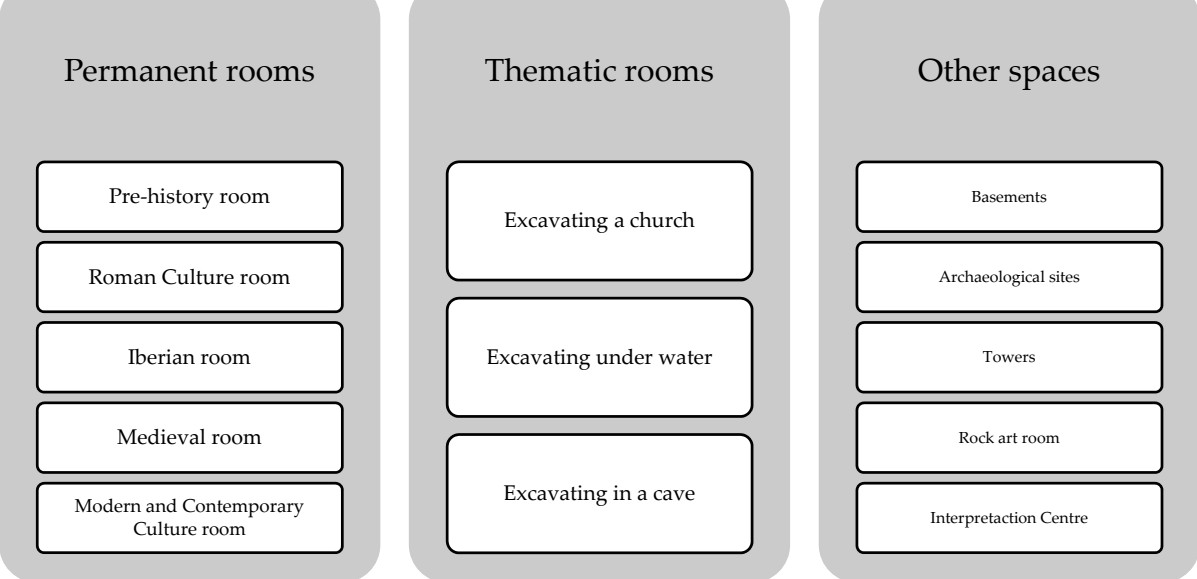

**Figure 2.** Spatial organization of the Alicante museum. Source: authors' own work.

### 2.3.3. Participants in the Research

The sampling technique used for the selection of the participants was a mixed procedure (cluster sampling). The participating subjects were the cultural managers (museum educators, guides and the education managers of both museums) and early years, primary education, and compulsory secondary education teachers who, throughout 2017 and 2018, visited the archaeological museums with their class groups in order to carry out a visit, guided tour, workshop or any other activity in the museum.

The participating teachers of the MU ($n$ = 160) and the MA ($n$ = 272) present an average age of around 43. Their average experience in the teaching profession is 17 years.

The school groups which visited the museums most were from primary education (50.8%), followed by early years groups (27.6%). The fewest number of visits were those corresponding to the stage of compulsory secondary education, representing 21.6% of the total (Figure 3).

**Total percentage**

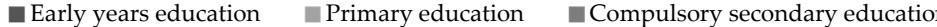

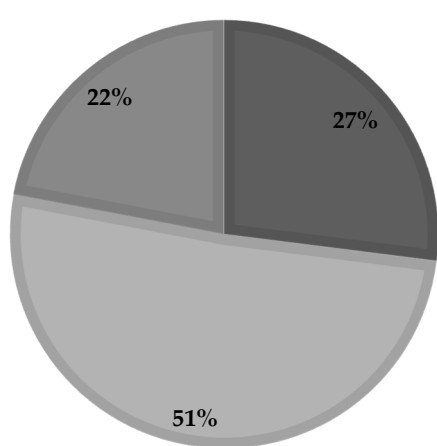

**Figure 3.** Total percentage of school groups visiting the archaeological museums according to their educational stage. Source: authors' own work.

The average age of the educators was 36 years of age, with an average professional experience of eight years. Of the total of educators surveyed (*n* = 18), when differentiated according to museum (*n*MU = 4; *n*MA = 14), the average teaching experience does not fluctuate. Although the educational staff of the MU are significantly fewer and are not hired by the museum, but are subcontracted via an external company, they are permanent workers of the museum and have an average professional experience of 8.5 years. On the other hand, the group of educators of the MA have less average experience (8.21 years). Most educators from the MA, those responsible for carrying out guided tours, workshops, and other activities with school children, are not hired directly by the museum but are subcontracted from an external company and most of them have temporary contracts. The permanent staff of the museum have an average professional experience of more than ten years.

*2.4. Survey Structure*

The tool used for the data collection was the MUSELA© questionnaire in its two versions (aimed at teachers -MUSELA DO- and museum educators -MUSELA EDU-), elaborated ad hoc and validated via a study which was both exploratory and confirmatory [37].

The aim of both tools is to analyze the relationship and lines of collaboration which are established between museums and schools and to verify whether archaeological museums are useful tools which offer educational strategies to teachers to be used in their planning in the teaching of the social sciences in general and history in particular.

The construction process of the questionnaires was forged from the main theoretical principles regarding the educational potential of museums [38–41]), the importance of the educational agents who intervene in the interaction between heritage and education [42], models of collaboration which are established between museums and schools [43] and the need for sequential planning of school visits to museums [15,44,45].

The research design employed for the construction and validation of the MUSELA© questionnaires is based on the classical test of theory (CTT). The second version was submitted to the expert judgement procedure with the aim of analysing the degree of agreement between the value judgments emitted by the participants of the group of external judges. The Kendall's W coefficient of concordance has made it possible to verify that there is agreement among the judges who intervened in the validation process of the MUSELA© tool by way of the scale designed for this purpose. Furthermore, the internal consistency and validity of the construct of the MUSELA questionnaires was analyzed via

pilot and confirmatory studies. The analysis of the reliability of the pilot tool aimed at teachers via the Cronbach's alpha model (total scale and two halves), both for the items as a whole and for the grouping of the items by dimensions, reveals that the scale is appropriate. The analysis of the main components reduces the 38 variables to seven factors which explain almost 99% of the total variance. The MUSELA EDU© validation process (in the pilot study) has led to substantial improvements in the tool, in addition to verifying its reliability. The results of the statistical tests indicate that the MUSELA EDU© tool has improved in terms of internal consistency, reaching a value of two points higher than the tool in its pilot study ($\sigma = 0.896$). Moreover, despite the fact that the variance explained in the definitive tool is lower, it presents a high degree of variability (100% in the pilot study, $\sigma = 0.65$ y 94.59% in the confirmatory study, $\sigma = 0.90$).

The results of the confirmatory factor analysis (CFA) indicate that the MUSELA DO© questionnaire presents a good internal consistency ($\sigma = 0.832$). It was not necessary to eliminate any variables from the total scale. The Exploratory Factor Analysis developed at MUSELA DO© is relevant due to its sampling adequacy, as demonstrated by the results of the Kaiser–Meyer–Olkin (KMO) test, which oscillates at a value of 0.720.

El MUSELA EDU© has a good internal consistency ($\sigma = 0.896$). Therefore, the items of the total scale of the instrument and of each of the scales of the dimensions enjoy internal consistency. It was not possible to calculate the KMO tests to determine the construct validity due to the small sample size of the study from the MUSELA EDU©. The questionnaire has content validity, although this content does not exactly correspond to the dimensions of the questionnaire.

The final version of the study presents a body of questions organized into five blocks: general information, the teacher's or educator's opinion regarding museums and school visits, the collaboration between schools and archaeological museums, planning visits to archaeological museums and the overall evaluation of the visits to the archaeological museums. The MUSELA DO© questionnaire consists of 27 questions and the MUSELA EDU© of 24 questions.

The questionnaire was self-administered by each participant individually. It was delivered in person to teachers visiting the participating archaeological museums with their groups of students in order to be answered throughout the duration of the school visit. The questionnaire was also given to the educators of these museums. As Monroy, González-Geraldo and Hernández-Pina [46] point out, the aim is for the participants to return the questionnaire answered in full. Thus, the way in which the questionnaires were administered implied a great cost in terms of their distribution and collection.

The study focuses on the data collected from the section on planning the field trip to the archaeological museum, specifically on those variables which make reference to the design, development and follow-up of the museum visit by the educational agents during the school visit (dichotomous, ordinal scale, various response options and multi-response). The questions from the questionnaire which have been taken into account for this study can be observed in Appendix A (the variables used by the teachers) and Appendix B (the variables which gather the information from the museum educators).

### 2.5. Procedure for the Collection, Handling and Analysis of the Data

After the data was collected, data matrices were elaborated using the Statistical Package for the Social Sciences IBM® SPSS (version 24). After executing the program and creating the data matrices, these were explored and the categorized data were systematized into different variables both according to their nature or measurement scale and their methodological function in the objectives and hypothesis of this research.

The analytical techniques were systematized according to the purpose for which they had been applied: descriptive statistics, including the cross tables or contingency tables. Graphic techniques, bar charts, histograms, sector diagrams, etc. were used as analytical strategies, and were developed with the statistical program IBM SPSS for Windows (version 24) and with Microsoft Excel for Mac (version 16.24).

In addition, the Chi-squared statistical test was used to show statistically significant associations between different types of activities or resources used by teachers and museum educators. In order to contrast the hypotheses, non-parametric tests for two independent samples were used to compare ranges (Mann–Whitney U test). For all of these cases, bilateral statistical significance tests were used, assuming 0.05 as the level of significance or critical level.

## 3. Results

### *3.1. Design and Development of Activities*

With the aim of comparing the activities prepared in the design of the field trips from the perspective of teachers and museum educators, the frequency with which museum educators carry out different types of activities was verified. From the point of view of the teachers, the aim was to determine what activities they prepare and in which educational space (museum or classroom) they are carried out.

In the design adopted, the following were presented as independent variables: the role of the educational agents (divided into two analysis variables: teachers and museum educators); the educational space (the museum or the classroom); and the two educational museum spaces. As a dependent variable, the type of activities proposed was taken into account. These were, in turn, categorized into eleven analysis variables, taking into account the types of activities proposed in the teaching of current and historical societies [33]: experimentation or artistic workshops; role plays or theatrical representations; cooperative tasks for problem-solving; treasure hunts; debates; brainstorming; round table discussions; observation tasks and/or handling of objects; fieldwork; data representation; interviews with experts or family members; viewing of audio-visual materials; the creation of a classroom museum or exhibition of objects.

### 3.1.1. Results as Far as the Role of the Educational Agents Is Concerned

(a)   Development of activities from the point of view of the teachers:

From a general perspective, six out of ten teachers stated that they do indeed prepare some activities within the design of the field trip to the archaeological museum. On the other hand, four out of ten stated that they do not prepare any activities (Figure 4).

Within this 60% of teachers who prepare activities, the most used types of activities (Figure 5) are experimentation or artistic workshops (44.5% of the total of teachers), debates, brainstorming or round table discussions (32.2% of the total), observation tasks and/or handling of objects (21.8% of the total), the viewing of audio-visual materials (45.9% of the total) and the creation of a classroom museum or exhibition of objects (15.5% of the total).

As it can be observed in the results of the total percentages of the teachers (Table 3) who use each of the activities (the total percentages presented are not accumulated percentages), none of them is used by at least 50% of the total of participating teachers ($n = 433$).

Among the most used activities in the classroom are the viewing of audio-visual materials (34.2%), debates, brainstorming or round table discussions (28.6%) and the creation of a classroom museum or an exhibition of objects (15%).

According to the teachers, the activities which are carried out more in the museum space are experimentation or artistic workshops (31.4%) and observation tasks and/or handling of objects (12.4%), followed by the viewing of activities (11.7%).

(b)   Development of pre-visit activities and types of activities carried out beforehand within the design of the field trip according to the teachers:

The aim was to determine whether there were any statistically significant differences between the frequency with which pre-visit activities were carried out and the type of activities that teachers state that they carry out before a visit.

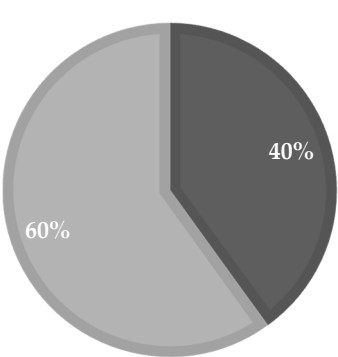

**Figure 4.** Valid percentages of teachers who prepare activities within the design of the school field trip. Source: authors' own work.

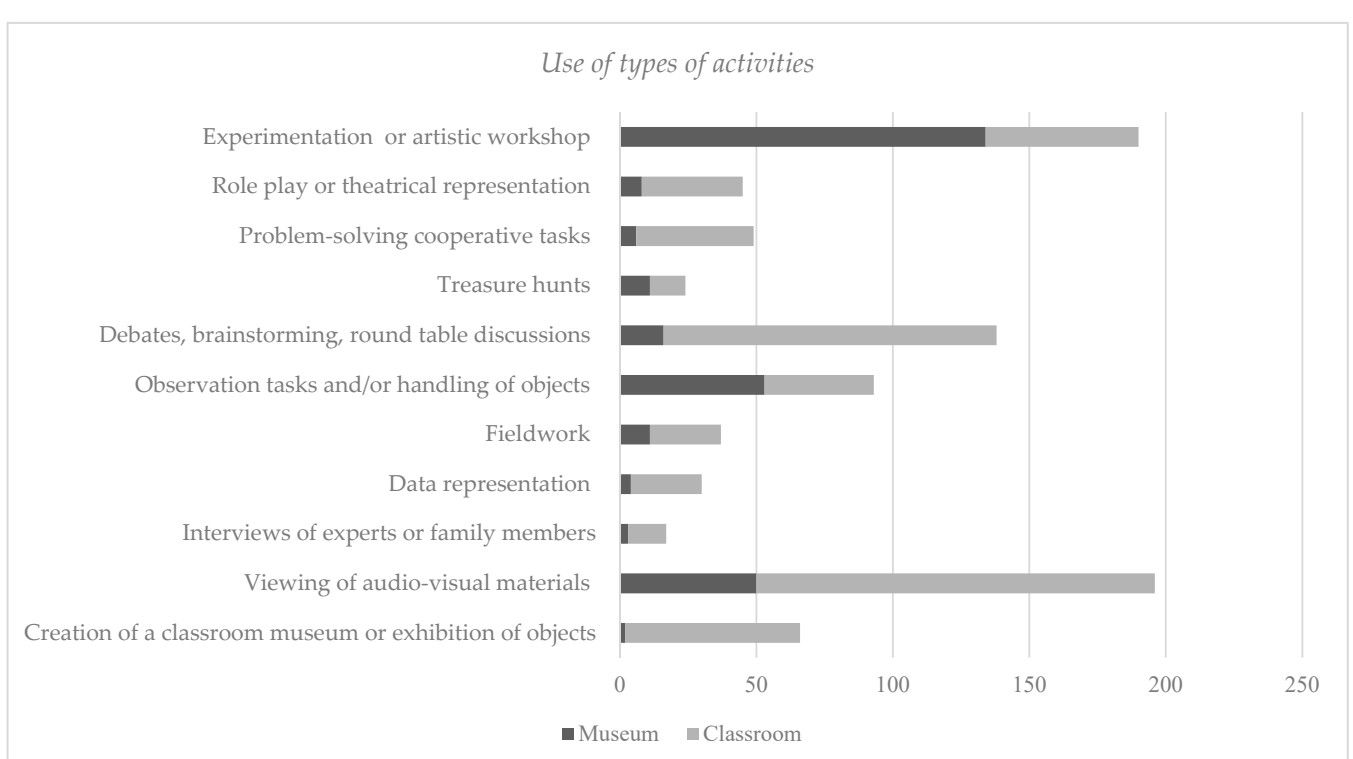

**Figure 5.** Frequency of the use of the types of activities and the space in which they are carried out according to teachers. Source: authors' own work.

The application of the nonparametric Mann–Whitney U test for all the analysis variables categorized according to types of activity determines that the difference between the types of activities carried out prior to the visit and the development of prior activities is statistically significant for all of the cases ($p < 0.05$), with the exception of the "Treasure hunt" activity ($p > 0.05$).

**Table 3.** Table of the frequencies and total percentages (not accumulated) of the use made of each of the types of activities proposed according to the teachers.

| Educative Space Type of Activity | Museum | Classroom |
|---|---|---|
| | Total Percent | |
| Experimentation or artistic workshop | 31.4 | 13.1 |
| Role play or theatrical representation | 1.9 | 8.7 |
| Problem-solving cooperative tasks | 1.4 | 10.1 |
| Treasure hunts | 2.6 | 3 |
| Debates, brainstorming, round table discussions | 3.7 | 28.6 |
| Observation tasks and/or handling of objects | 12.4 | 9.4 |
| Fieldwork | 2.6 | 6.1 |
| Data representation | 0.9 | 6.1 |
| Interviews of experts or family members | 0.7 | 3.3 |
| Viewing of audio-visual materials | 11.7 | 34.2 |
| Creation of a classroom museum or exhibition of objects | 0.5 | 15 |

Source: authors' own work.

The results of the average ranges of each of the activities which present statistically significant differences with carrying out prior activities indicate that these differences develop in favor of the teachers who carry out such prior activities ($PR_{Yes\ Previous\ activities}$ > $PR_{No\ Previous\ activities}$). In other words, the frequency with which prior activities are carried out is greater in those types of activities which are carried out before the visit.

(c) Development of post-visit activities and those types of activities which are carried out subsequently within the design of the field trip according to the teachers:

The aim of this section is to determine whether there are statistically significant differences between the frequency with which the school groups carry out post-visit activities and the types of activities carried out after the field trip to the archaeological museum.

The results of the application of the nonparametric Mann–Whitney U test determine that the differences of all of the activities which are carried out after the visit and the frequency with which activities are carried out after the visit are statistically significant ($p < 0.05$), with the exception of the activities "Role play or theatrical representation", "Treasure hunts" and "Data representation", which do not present statistically significant differences ($p > 0.05$).

Therefore, these differences cannot be put down to chance. Rather, the average ranges between the types of activities which are carried out after a visit and those which are not are different from the statistical point of view. More specifically, for the analysis variables, the results of the average ranges indicate that the range is greater for those activities which are carried out after a visit and lesser for those which are not ($PR_{Yes\ Post-visit\ activities}$ > $PR_{No\ Post-visit\ activities}$). In this way, it is possible to understand that the frequency with which post-visit activities are carried out is greater in those types of activities which are carried out after a visit.

(d) Development of activities from the point of view of the museum educators:

A total of 94.4% of the museum educators stated that they carry out some type of activity during school visits. The average frequency with each of the activities is used within the types proposed is reflected in Figure 6.

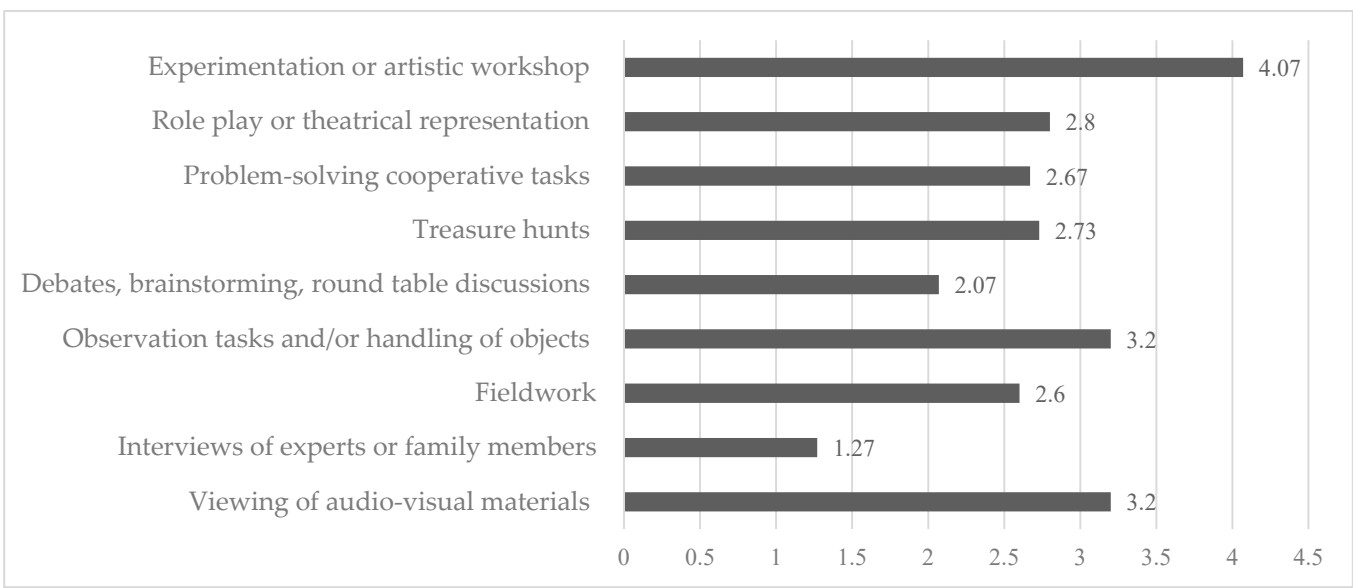

**Figure 6.** Average values on the frequency of the use of the different types of activities according to the museum educators. Source: authors' own work.

Experimentation or artistic workshops are the most frequently carried out activities, given that the results of the average values indicate that they are implemented extremely frequently. They are followed by the viewing of audio-visual materials and the observation and/or handling of objects, which are carried out with a certain degree of frequency. The remaining activities are carried out occasionally, rarely or hardly ever, as is the case of interviews of experts or family members.

As can be observed in Figure 7, the most frequently carried out activity is always the experimentation or artistic workshop (35.3% of the total), 47% of the museum educators state that they employ this activity with great frequency.

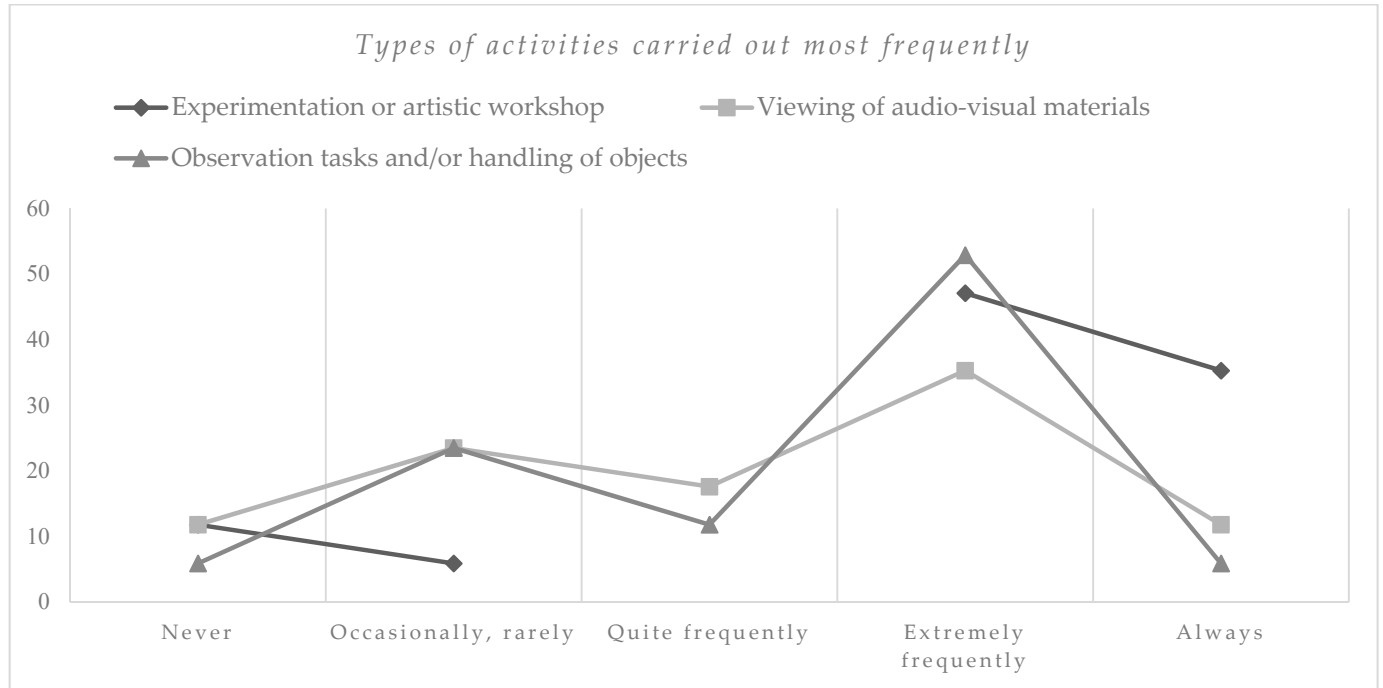

**Figure 7.** Valid percentages of the frequency of the use of the different types of activities most used from the perspective of the museum educators. Source: authors' own work.

The observation and/or handling of objects is carried out quite frequently according to 53% of the museum educators surveyed. Only 6% stated that this activity is always carried out, whereas 23.5% stated that it is carried out occasionally or rarely.

Audio-visual materials are shown frequently according to 35.3% of the educators, with 12% claiming to show them always and 17.6% quite frequently.

The rest of the activities (Figure 8) are carried out less frequently. A total of 75% of the museum educators consider that interviews of experts or family members are never implemented or only occasionally or rarely according to 25% of those surveyed.

*Types of activities carried out with less frequency*

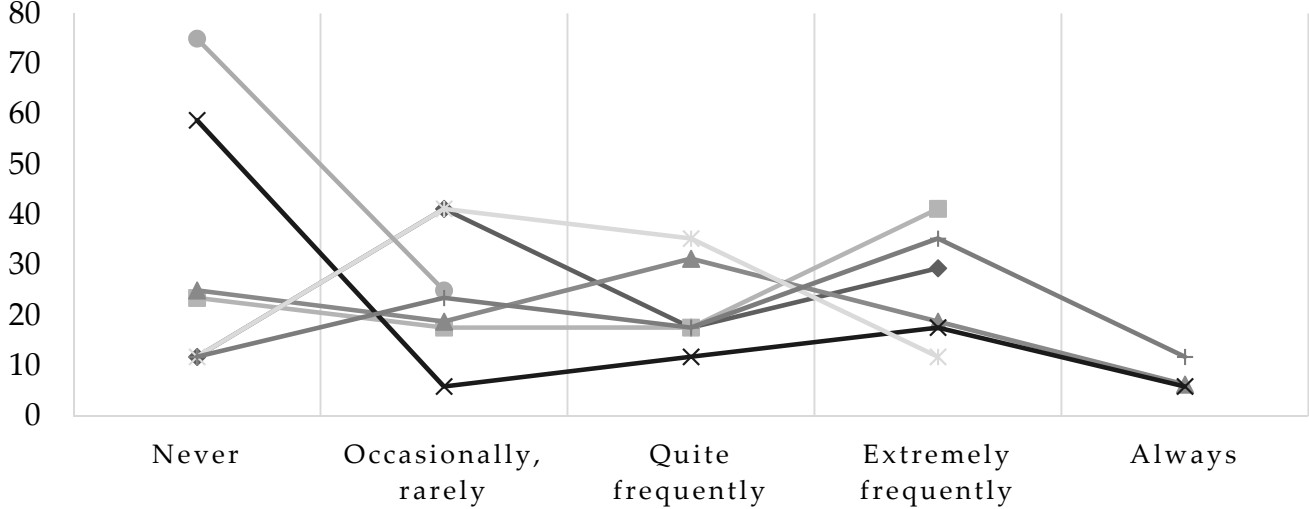

**Figure 8.** Valid percentages of the frequency of the use of the different types of activities used least from the perspective of the museum educators. Source: authors' own work.

Debates, brainstorming or round table discussions are also not carried out very frequently (only 17.6% stated that they use these activities), 59% stated that they never put these activities into practice.

Role playing and activities relating to fieldwork are occasionally or rarely carried out according to 41% of the museum educators surveyed.

3.1.2. Results Regarding the Educational Space

(a)    Results from the teachers' point of view:

It was also considered to be of interest to verify whether there were statistically significant differences between the type of activities used by the teachers and the educational space in which these tasks were carried out.

The results of the Chi-squared statistical test showed statistically significant associations between the educational space and the development of the field trip as fieldwork

($p < 0.05$; $\chi^2 = 18.098$) and the viewing of audio-visual materials as a type of activity ($p < 0.05$; $\chi^2 = 4.799$).

The statistically significant differences of the activity on the fieldwork arise in favor of those teachers who do not use them in the classroom or in the museum (92.3% of the total of teachers participating in the research (Table 4).

**Table 4.** Total percentages of school groups which carry out the activity on fieldwork in the museum and at school.

| Use | | Classroom Space | | |
| --- | --- | --- | --- | --- |
| | | No | Yes | Total |
| Museum space | No | 92.3% | 5.2% | 97.4% |
| | Yes | 1.6% | 0.9% | 2.6% |
| Total | | 93.9% | 6.1% | 100.0% |

Source: authors' own work.

Likewise, the statistically significant differences of the activity relating to the viewing of audio-visual materials is in favor of those who do not use this resource, be it in the museum or in the classroom, corresponding to 59.7% of the total of participating teachers (Table 5).

**Table 5.** Total percentages of school groups carrying out the activity of viewing audio-visual materials in the museum and the classroom.

| Use | | Classroom Space | | |
| --- | --- | --- | --- | --- |
| | | No | Yes | Total |
| Museum space | No | 59.7% | 28.6% | 88.3% |
| | Yes | 6.1% | 5.6% | 11.7% |
| Total | | 65.8% | 34.2% | 100.0% |

Source: authors' own work.

(b)   Results from the point of view of the educators

The differences between the frequency of use of each of the types of activities carried out by the educators (dependent variable or criterion) and the educational and museological space in which they were carried out (independent variable or predictor) were analyzed. The results of the nonparametric statistical test for independent $K$ samples, the Mann–Whitney U test, determine that there are no statistical differences between the frequency of use of each of the types of activities carried out by the educators and the museological space in which they are carried out ($p > 0.05$).

Therefore, there is nothing to oppose the rejection of the alternative hypothesis and the acceptance of the null hypothesis:

**Hypothesis 1 (H1).** There are no differences between the frequency of use of each of the types of activities carried out by the educators and the educational museological space in which they are developed.

**Hypothesis 2 (H2).** There are differences between the frequency of use of each of the types of activities carried out by the educators and the educational museological space in which they are developed.

3.1.3. Results According to Educational Level

Taking into account the results according to educational level, Table 6 shows the total percentages of the use according to level for each of the activities (analysis variables).

Among the most used activities proposed in the previous sections of the study, experimentation or artistic workshops were the most used (16.9%) in the stage of primary education within the museum space during the school visit (16%), followed by early years education (10.3%), also in the museum and during the guided visit (9.6%). The activity of debate, brainstorming and round table discussions was mainly carried out in the primary classroom (17.1%) before the visit to the museum (7.3%). The viewing of audio-visual materials was also carried out to a great degree in the primary classroom (20.4%) before the school visit to the museum (17.8%).

**Table 6.** Total percentages of the use of activities according to educational level.

| Type of Activity | Educational Stage | Museum | Classroom | Before | During | After |
|---|---|---|---|---|---|---|
| Experimentation or artistic workshop | Early years | 10.3% | 6.6% | 5.6% | 9.6% | 4.2% |
| | Primary education | 16.9% | 5.2% | 3% | 16% | 3.7% |
| | Secondary education | 4.2% | 1.4% | 0.7% | 4.2% | 0.7% |
| Role Play or theatrical representation | Early years | 0.7% | 4.4% | 2.8% | 0.7% | 1.6% |
| | Primary education | 0.7% | 3.3% | 2.3% | 0.7% | 0.9% |
| | Secondary education | 0.5% | 0.9% | 0.5% | 0.5% | 0.5% |
| Problem-solving cooperative tasks | Early years | 0% | 2.8% | 2.6% | 0.5% | 0.7% |
| | Primary education | 0.9% | 5.6% | 3.5% | 0.9% | 2.6% |
| | Secondary education | 0.5% | 1.6% | 1.2% | 0.5% | 0.7% |
| Treasure hunts | Early years | 1.2% | 0.9% | 0.7% | 1.2% | 0.5% |
| | Primary education | 1.2% | 1.9% | 1.2% | 0.9% | 0.9% |
| | Secondary education | 0.2% | 0.2% | 0% | 0.2% | 0.2% |
| Debate, brainstorming, round table discussions | Early years | 1.6% | 6.3% | 4% | 1.9% | 4% |
| | Primary education | 1.6% | 17.1% | 7.3% | 1.9% | 12% |
| | Secondary education | 0.5% | 5.2% | 2.1% | 0.7% | 4.7% |
| Observation tasks and/or handling of objects | Early years | 4.2% | 3.5% | 2.8% | 4.2% | 1.4% |
| | Primary education | 6.3% | 4.9% | 2.8% | 6.1% | 2.6% |
| | Secondary education | 1.9% | 0.9% | 0.9% | 1.9% | 0.2% |
| Fieldwork | Early years | 0.7% | 1.2% | 1.2% | 0.5% | 0.5% |
| | Primary education | 1.4% | 3.5% | 2.8% | 1.4% | 0.9% |
| | Secondary education | 0.5% | 1.4% | 0.7% | 0.5% | 0.9% |
| Data representation | Early years | 0.2% | 2.1% | 1.2% | 0.2% | 1.2% |
| | Primary education | 0.7% | 2.3% | 1.9% | 0.7% | 0.7% |
| | Secondary education | 0% | 1.6% | 0.5% | 0% | 1.2% |
| Interviews of experts or family members | Early years | 0% | 1.4% | 1.2% | 0% | 0.2% |
| | Primary education | 0.7% | 1.2% | 1.2% | 0.5% | 0.5% |
| | Secondary education | 0% | 0.7% | 0.7% | 0% | 0.5% |
| Viewing of audio-visual materials | Early years | 3.7% | 8.2% | 7.3% | 4.2% | 3.3% |
| | Primary education | 7.3% | 20.4% | 17.8% | 7.7% | 7% |
| | Secondary education | 0.7% | 5.6% | 4.9% | 0.7% | 2.8% |
| Creation of a classroom museums or exhibition of objects | Early years | 0% | 6.1% | 3.5% | 0.5% | 2.6% |
| | Primary education | 0.5% | 6.6% | 3.5% | 0.9% | 3.7% |
| | Secondary education | 0% | 2.3% | 0.7% | 0% | 1.9% |

Source: authors' own work.

The Pearson's Chi-squared statistical test applied to the dependent or predictor variables according to the stage of education (early years, primary and secondary) which acts as the independent or criterion variable, determines that there are significant associations in the four types of activities which appear in Table 7 (experimentation or artistic workshop, role play or theatrical representation, viewing of audio-visual materials and creation of a classroom museums or exhibition of objects). These associations occur in the stages of early years and primary education and never in the stage of compulsory secondary education.

**Table 7.** Significant associations in the activities employed according to educational stages.

| Educational Stage | Early Years | | | | | Primary Education | | | | |
|---|---|---|---|---|---|---|---|---|---|---|
| Type of Activity | Museum | Classroom | Before | During | After | Museum | Classroom | Before | During | After |
| Experimentation or artistic workshop | | X | X | | X | X | | | | |
| Role Play or theatrical representation | | X | X | | | | | | | |
| Viewing of audio-visual materials | | | | | | X | | | X | |
| Creation of a classroom museums or exhibition of objects | | | | | | | X | X | | |

NOTA: X = $p < 0.05$; Source: authors' own work.

There are significant associations ($p < 0.05$) in the activity experimentation or artistic workshop in favor of the early years stage in the activity carried out in the classroom and before and after the school visit to the museum. There are also significant associations in the workshops carried out in the museum ($p < 0.05$) in favor of the elementary stage of education.

The activity Role Play or theatrical representation also presents a significant association with an associated probability of $p < 0.05$ for activities carried out in the classroom before the visit and in favor of the early years stage.

The significant associations produced in the activities regarding Viewing of audio-visual materials in the museum and during the visit are in favor of the primary stage. Last of all, the activity on the Creation of a classroom museums or exhibition of objects is also in favor of the primary groups, although, in this case, the significant association occurs in the activity which is carried out in the classroom, prior to the school visit to the archaeological museum

### 3.2. Use of Materials and Resources

Last of all, we aim to describe the use of the resources and materials from the point of view of the educational agents. In order to achieve this, the role of the agents (teachers and museum educators) and the educational space (museum and classroom) were taken into account as independent variables. The type of resources used was taken as the dependent variable, which has been categorized into seven analysis variables corresponding to the types of resources used:

1. Web pages, audio-visual videos, photographs or any ICT resources which do not imply the direct participation of the user. These are the educational resources employed, which, according to Cuenca, Molina and Martín, used for communication in the heritage context, fit into the classification of 'Passive ICT resources' [47].

2. Videogames, web applications, QR codes, augmented reality, virtual museums or any ICT tools which do foster the active involvement of the user. These resources, within

the classification established for the communication of heritage, are known as 'Active ICT resources' [47].

3. Books, interviews, people from the immediate environment, etc., materials which act as sources of information, be it textual, oral, etc.
4. Worksheets or edited bibliographic material.
5. Objects or material remains which hold relevant historical information susceptible to analysis and study.
6. Models or reproductions.
7. Cartographic representations, maps.

3.2.1. Results Regarding the Role of the Educational Agents

(a)   Use of resources from the teachers' point of view

First of all, taking a general perspective of the results from the teachers' point of view, the use of resources compared to that of activities is 120% higher. Seven out of ten teachers consider that they make use of some materials or resources to complement the field trip, whereas only three out of ten state that they do not make use of any type of resource or material (Figure 9).

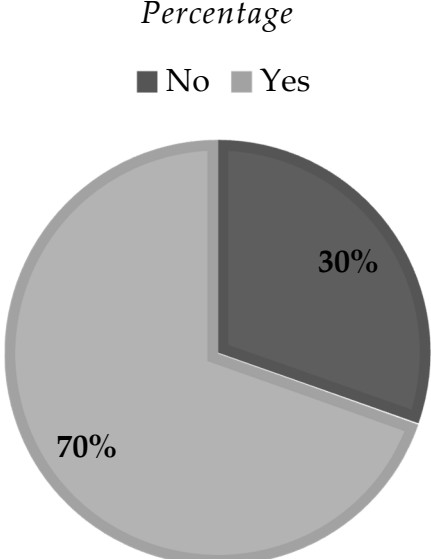

**Figure 9.** Valid percentages of the use of materials or resources to compliment field trips to archaeological museums according to teachers. Source: authors' own work.

In Table 8, it can be observed that very few teachers use or are aware of the use of these types of resources within the museum. On the other hand, in the classroom, the total percentages of teachers who use each of these types of resources are greater (the total percentages presented are not accumulated percentages).

Figure 10 shows the types of activities which are used most. In this regard, the use of ICT resources which do not imply the direct participation of pupils (the so-called 'Passive ICT resources') stand out. These resources are used in the classroom by 60.7% of the teachers. This could be compared with the teachers who stated that they use the viewing of audio-visual materials in the classroom (34.2% of the total). 46% of the teachers state that they use worksheets or other edited bibliographic materials.

As can be observed in Figure 12, the resources which museum educators claim to use with greater frequency during school visits are object sources and models or reproductions, which are always used by almost 60% of the participating educators. 'Passive ICT resources' are used extremely frequently by 53% of the educators, of whom 60% claim to use cartographic representations or maps with the same degree of frequency. All of these

resources are used with a certain degree of frequency by 12% of the educators and never by 6%.

**Table 8.** Frequencies and total percentages (not accumulated) of the use made of each of the resources proposed according to the teachers.

| Educative Space Type of Activity | Museum | Classroom |
|---|---|---|
| | Total Percent | |
| Passive ICT | 11.5 | 60.7 |
| Active ICT | 1.9 | 10.1 |
| Books, interviews, people from the immediate environment | 2.8 | 25.3 |
| Worksheets | 4.7 | 45.9 |
| Objects and material remains | 8.7 | 10.1 |
| Models or reproductions | 6.8 | 7 |
| Cartographic representations, maps | 3.5 | 15.9 |

Source: authors' own work.

*Use of types of resources*

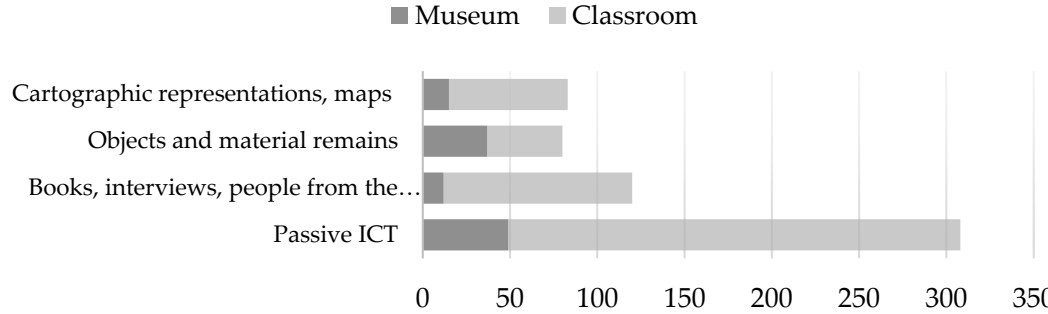

**Figure 10.** Frequency of the use of the types of activities and the space in which they are carried out according to the teachers. Source: authors' own work.

The most commonly-used resources in the museum are 'Passive ICT resources' such as videos, websites, photographs, etc. (11.5% of the total of the teachers) and objects or material remains, although less than 10% of the teachers claim to use the latter.

On the other hand, the least-used resources in the museum are the ICT resources which imply the participation of the pupils (the so-called 'Active ITC resources'), with only 2% of the teachers stating that they use them. Likewise, books, bibliographic and oral sources are used in the museum by only 3% of the teachers.

(b)    Use of resources from the point of view of the museum educators:

94.4% of the museum educators claim that they use some of these resources during school visits. As can be observed in Figure 11, which indicates the average values of each of the resources, those most frequently used by museum educators are 'passive ICT resources' and objects and material remains. The educators state that these are used extremely frequently, followed by models or reproductions and cartographic representations or maps, which they use with a certain degree of frequency. The remaining resources ('Active ICT resources', books, interviews, people from the immediate environment and worksheets) are used occasionally or rarely.

On the other hand, the resources used least frequently during school visits to the archaeological museum are written or oral sources, which, according to 60% of the educators, are never used, followed by 'Active ICT resources', which are never used by 54% of the educators. Furthermore, worksheets are used extremely frequently by 23.5% of the

educators, with a certain degree of frequency by 29.5%, occasionally by 12%, and never by 35.3% (Figure 13).

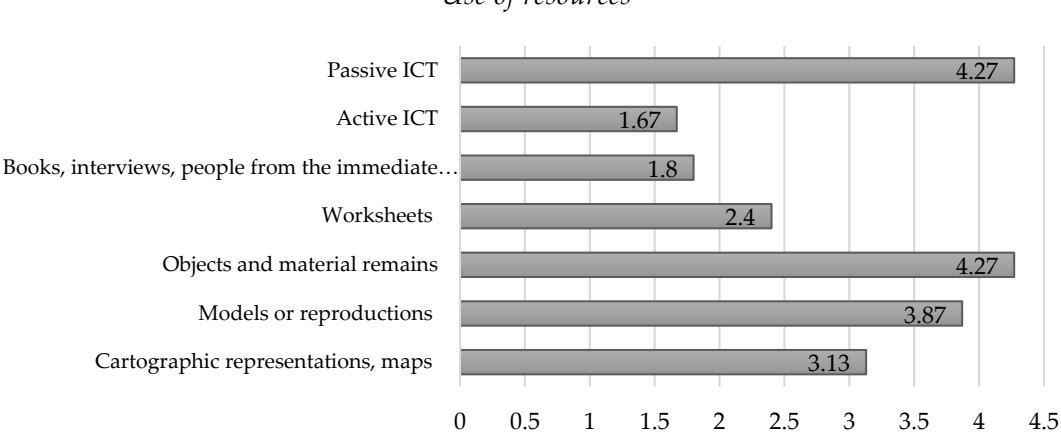

**Figure 11.** Average values regarding the frequency of use of the different types of resources according to the museum educators. Source: authors' own work.

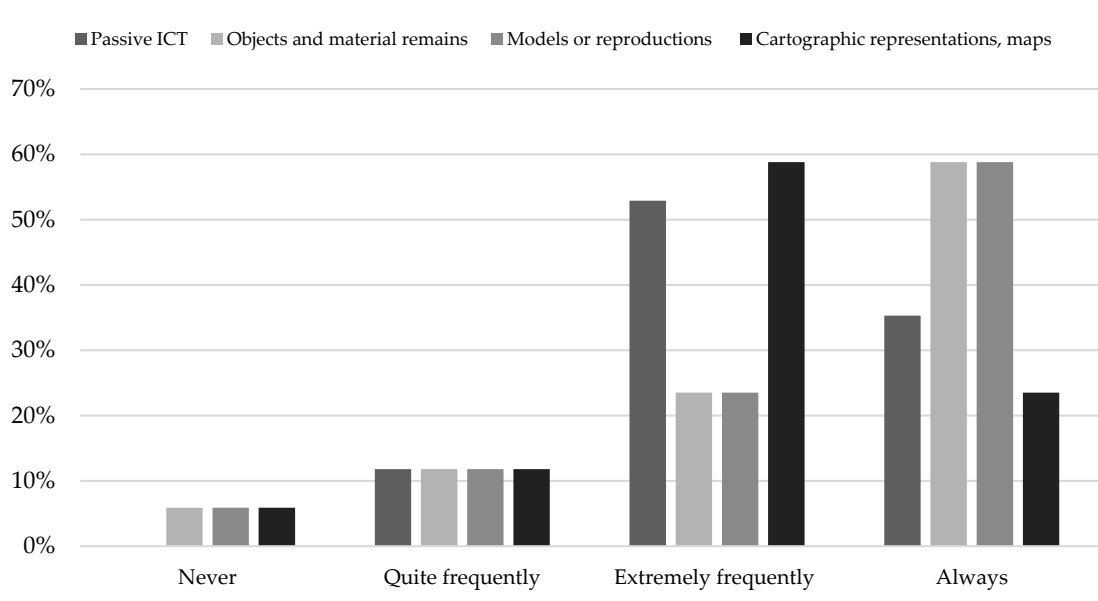

**Figure 12.** Valid percentages of the frequency of use of the different types of resources most commonly used from the perspective of the museum educators. Source: authors' own work.

3.2.2. Results Relating to the Educational Museological Space

(a)    Results from the teachers' point of view:

As was the case with the previous objective, statistically significant differences were sought between the type of resource used according to the teachers and the educational space in which the said resource was used. In order to achieve this, the non-parametric Chi-squared statistical test was applied.

The results showed statistically significant associations in two of the resources used: "active ICT resources" and cartographic representations or maps. As can be seen in Tables 9 and 10, the differences are in favor of the teachers who do not use these resources in the classroom or in the museum.

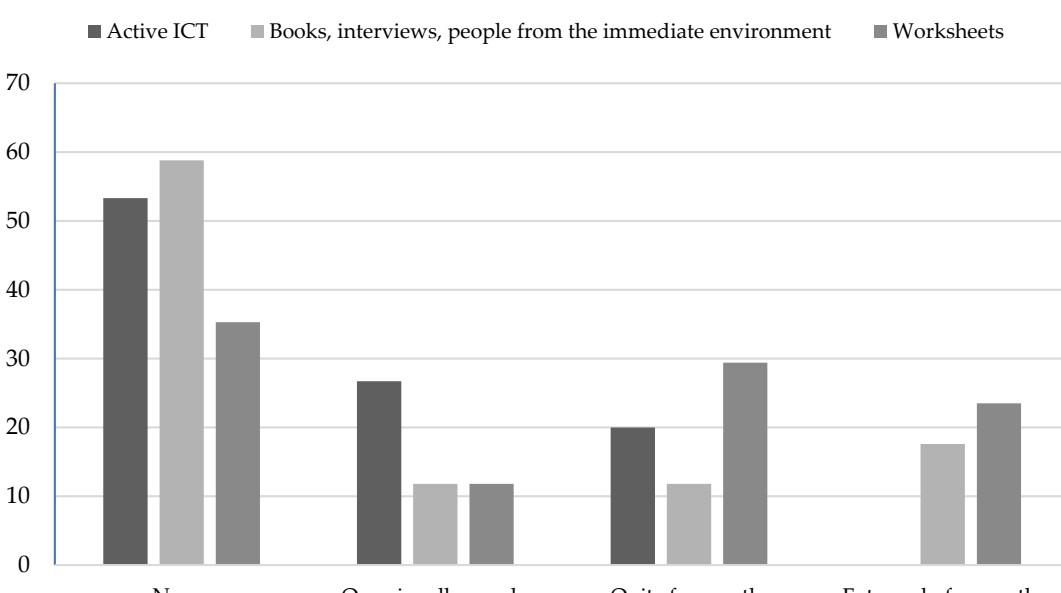

**Figure 13.** Valid percentages of the frequency of the use of the different types of resources used least from the perspective of the museum educators. Source: authors' own work.

**Table 9.** Total percentages of school groups using 'Active ICT resources' in the museum and at school.

| Use | | Classroom | | |
|---|---|---|---|---|
| | | **No** | **Yes** | **Total** |
| Museum | No | 89% | 9.1% | 98.1% |
| | Yes | 0.9% | 0.9% | 1.9% |
| Total | | 89.9% | 10.1% | 100.0% |

Source: authors' own work.

**Table 10.** Total percentages of school groups using the resources 'cartographic representations, maps' in the museum and at school.

| Use | | Classroom | | |
|---|---|---|---|---|
| | | **No** | **Yes** | **Total** |
| Museum | No | 82.2% | 14.3% | 96.5% |
| | Yes | 1.9% | 1.6% | 3.5% |
| Total | | 84.1% | 15.9% | 100.0% |

Source: authors' own work.

(b)    Results from the museum educators' point of view:

Last of all, it was analyzed whether there were any differences between the frequency of use of each of the materials and resources employed by the educators (dependent variable or criterion) and the educational museological space in which they are carried out (independent variable or predictor). The results of the nonparametric Mann–Whitney U test for independent K samples determine that there are no statistically significant differences between the frequency of use of each of the materials or resources used by the educators and the museological space in which they are carried out ($p < 0.05$). Therefore, there is nothing to oppose the rejection of the alternative hypothesis and the acceptance of the null hypothesis:

**Hypothesis 3 (H3).** There are no statistically significant differences between the frequency of use of each of the materials and resources used by the educators and the educational museological space in which they are carried out.

**Hypothesis 4 (H4).** There are statistically significant differences between the frequency of use of each of the materials and resources used by the educators and the educational museological space in which they are carried out.

3.2.3. Results According to Educational Level

As far as the results according to educational stage are concerned, Table 11 shows the total percentages of use according to level for each of the resources used (analysis variables). Among the most used resources proposed in the previous sections of the study, the ICT resources considered to be passive (web pages, audio-visual materials and photographs) are used to a greater degree in the primary classroom (34.7%) before the visit to the museum as a preparatory exercise (20.9%) and after the visit (23.9%). They are also used in the early years classroom, albeit to a lesser degree (14.8%). Books, interviews and people from the immediate environment are used by 11.7% of primary groups in the classroom. Worksheets are used both in the early years (15.7%) and in the primary classroom by 22%. In both groups they are used to a greater extent after the visit (12.2% by early years groups and 17.6% by primary groups).

**Table 11.** Total percentages of the use of resources according to educational level.

| Type of Resources | Educational Stage | Museum | Classroom | Before | During | After |
|---|---|---|---|---|---|---|
| Web pages, audio-visual materials, photographs | Early years | 3% | 14.8% | 7.7% | 4% | 8.5% |
| | Primary education | 6.6% | 34.7% | 20.9% | 5.9% | 23.9% |
| | Secondary education | 1.9% | 11.2% | 8.7% | 1.4% | 7.3% |
| Video games, apps, QR codes, augmented reality, virtual museum | Early years | 0.5% | 3% | 0.9% | 0.2% | 2.3% |
| | Primary education | 1.2% | 5.6% | 1.9% | 1.4% | 3.3% |
| | Secondary education | 0.2% | 1.4% | 1.4% | 0.2% | 0.7% |
| Books, interviews, people from the immediate environment, etc. | Early years | 0.9% | 9.1% | 6.3% | 1.6% | 5.6% |
| | Primary education | 1.9% | 11.7% | 6.1% | 2.6% | 7.5% |
| | Secondary education | 0% | 4.4% | 3.3% | 0% | 2.8% |
| Worksheets | Early years | 2.1% | 15.7% | 8.2% | 2.3% | 12.2% |
| | Primary education | 1.6% | 22% | 9.4% | 3% | 17.6% |
| | Secondary education | 0.9% | 8.2% | 4.4% | 0.7% | 5.4% |
| Objects and material remains | Early years | 2.1% | 6.1% | 4.4% | 2.1% | 4.7% |
| | Primary education | 4.9% | 3% | 2.1% | 4.4% | 2.1% |
| | Secondary education | 1.6% | 0.9% | 1.4% | 1.2% | 0.5% |
| Models or reproductions | Early years | 1.6% | 2.6% | 1.4% | 1.4% | 2.1% |
| | Primary education | 4.4% | 3.3% | 1.9% | 4.2% | 1.6% |
| | Secondary education | 0.7% | 1.2% | 0.9% | 0.7% | 0.7% |
| Cartographic representations, maps | Early years | 0.7% | 2.6% | 0.9% | 0.7% | 2.1% |
| | Primary education | 2.1% | 7.7% | 3.7% | 2.3% | 5.4% |
| | Secondary education | 0.7% | 5.6% | 4.2% | 0.5% | 3.5% |

Source: authors' own work.

Pearson's chi-square test of association was applied with an associated probability of $p < 0.05$. The results shown in Table 12 show that there are significant associations in

five of the seven types of resources analyzed. Specifically, the associations arise in passive ICT resources in favor of the primary stage, used in the classroom both before and after the museum visit. The use of Books, interviews, people from the immediate environment, etc. presents a significant association in favor of those used during the museum visit by early years groups. Worksheets present significant associations in favor of their use in the primary classroom and after the visit. The use of Objects and material remains is significantly associated in favor of their use in the early years classroom both during and after the visit and the resources related with Cartographic representations and maps present significant associations in favor of their use in the primary classroom and before the visit to the museum. The resources used by the groups from the stage of compulsory secondary education do not present significant associations ($p > 0.05$).

**Table 12.** Significant associations in the resources used according to educational stage.

| Educational Stage | Early Years | | | | | Primary Education | | | | |
|---|---|---|---|---|---|---|---|---|---|---|
| Type of Resources | Museum | Classroom | Before | During | After | Museum | Classroom | Before | During | After |
| Web pages, audio-visual materials, photographs | | | | | | | X | X | | X |
| Books, interviews, people from the immediate environment, etc. | | | X | | | | | | | |
| Worksheets | | | | | | | X | | | X |
| Objects and material remains | | X | X | | X | | | | | |
| Cartographic representations, maps | | | | | | | X | X | | |

NOTA: X = $p < 0.05$; Source: authors' own work.

## 4. Discussion and Conclusions

Regarding the design and implementation of activities and according to the results of the research, 60% of the groups currently carry out activities during guided visits to museums. The most frequent activities in museums are, according to the educational agents, experimentation and artistic workshops, followed by observation tasks and/or the handling of objects and viewing of audio-visual materials. This agrees to some extent with the conclusions of [32]. Within the museum's experimentation or artistic workshops, there are other workshops designed to develop an artistic activity, many of which seek to awaken interest in archaeological work, initiate students in archaeological retrospection work (via a small archeodrome) and to assess student learning through questionnaires. These workshops are integrated into the set of activities based on games as motivation, while the observation or handling of objects represents an activity that aims to research and analyze a specific topic under study. The viewing of audio-visual materials would give rise to an activity for the presentation or synthesis of information, before, during or after the visit. The purpose of the study by Ampartzaki et al. (2013) is "to integrate the ideas of community of practice and participatory action research by referring to the case study of a synergy between museum educators and academic researchers" [15] (p. 20). The answers to the questionnaire on the informal theoretical understanding reveal information about the nature of the activities carried out by the school groups and the resources used in each program. The main basic activity performed during the visit was limited to looking at the objects on display. To a lesser extent, there were activities requiring movement. On applying the educational program in its two versions, these basic activities became less frequent and there was a growing use of the following types of activity [15] (p. 15):

collaborative work in groups; looking at books, photographs, videos, etc.; handling objects; drawing and/or writing; digging. There were more hands-on activities and more observation of iconographic and textual resources, and these were accompanied by an increase in groupwork.

The results of the present study coincide to a certain degree with the work of Ampartzaki et al. (2013) [15]. The activities which make reference to watching audio-visual materials are those used most frequently in the classroom before the visit with the aim of preparing the activity. Ampartzaki et al. increased this type of activity in the last two versions of their program. In contrast, carrying out "problem-solving cooperative tasks" is practically inexistent in museums (1.9%) and in the classroom (8.7%) according to teachers. The same is true for "observation tasks and/or the handling of objects". In [15] the increase in these activities is highlighted. However, in our study their use by teachers visiting the two archaeological museums does not reach 7% of school groups and those who do use them state that they do so to a greater extent during the visit to the museum.

Teachers also provide us with information on the tasks they carry out in the classroom after the visit to the museum; this information does not appear in any other study on similar characteristics. Among the activities most used in the classroom are the viewing of audio-visual materials, debates and idea sharing. The use of audio-visuals in the classroom is methodologically related to activities focused on motivation or for the collection of information. Elsewhere, the debate is aimed at a reflection based on the information obtained during the visit. These activities do not require students to work with the scientific method (typical of research work within a learning project) but are complementary to the experiential visit.

About use of materials and resources, the results of the research show that from the perspective of the educational agents re-sponsible for the school visits, the resources most frequently used are videos and photographs, followed by objects and material remains and models or reproductions. These results point to a considerable use of resources that make the visit more worthwhile, according to [15]. According to the teachers, the resources used in the classroom to complement the visit include websites, videos and photographs, worksheets and books. As highlighted in both cases, traditional ICT resources that do not encourage the inter-action and participation of students in the museum space and in the classroom are the most used, while the least used are ICT resources that turn students into active subjects.

As regards the use of resources, Ampartzaki et al. (2013) [15] report that dioramas were used during the visit prior to implementing a program. This resource was used significantly less frequently after implementing the program, when the materials used were: special tools (spade, goggles, brush, clipboard, graph paper and pencil); sandpits; books; photographs; videos. These resources are used throughout the activities mentioned above. In the first version, only the special tools for fieldwork and sandboxes for archaeological excavation were used. In the second version, books, photographs and videos were incorporated. These results coincide with the information shown by teachers throughout our study on the use of books, which are mainly employed by groups from the primary stage of education. In contrast, the use of resources such as models, reproductions or objects is practically inexistent.

Based on the results formulated in this study and on the improvements applied by introducing new resources and activities included in the study by Ampartzaki et al. (2013) [15], it is clear that there is a need to be able to develop strategies for improving the educational use made by school groups of the resources and activities of the archaeological museums. In the introduction, the need was stressed to generate sustainable learning which would enable the educational use of resources and activities. The results of the research have shown that the use thereof is insufficient and is not proportional in the different stages of education.

The results presented here have shown the clear need to contribute towards museums providing a greater range of active ICT resources to facilitate learning among school

children during field trips in such a way as to compliment the visit before or after in the classroom. Among the aims of the research by [16] and [17] is the analysis of the perceptions of students between 14 and 16 years old of the educational actions carried out in museums. One of the results highlights the lack of interest shown by teenagers in archaeological museums. The main motive assumed by the authors is due to the poor use of museum resources (text panels, room sheets, illustrations and pieces in showcases). In the words of the authors on traditional museum resources: "these types of resources do not exactly create an atmosphere of dialogue between the museum and its users ( . . . ) the most important thing for a museum to adapt to this type of public is its predisposition to interact with them, to transform itself from a passive exhibition to an active laboratory" [38].

Twenty years after Hannon and Randolph [42] argued that educators were not focused on the use of technology in their programing, their contention remains valid, despite the heavy emphasis that was, and continues to be, placed on using technology.

A school trip to an archaeological museum in itself (while of great educational and didactic potential) will not guarantee the students' successful learning of history without planning, design and preparation, measuring the methodological parameters that are going to be developed: communication between the interested parties, interaction throughout the visit, carrying out pre-visit and post-visit activities, establishing learning goals and/or objectives, performing activities in the museum and the classroom, use of materials and resources, etc.

The study shows the activities and resources which are predominant in school field trips to museums. The conclusions drawn regarding the frequency of the various actions surrounding the planning of the visit are systematized in Figure 14 below.

**Figure 14.** Main conclusions on the frequency with which educators and teachers plan the school visit to the museum. Source: authors' own work.

In conclusion, the study shows that, according to the educational agents, activities are carried out before and after the school visit to the museum with some frequency:

- The educators prepare activities prior to the visit with the teachers or continue in contact with the schools for later work on occasions. Eight out of ten educators consider that they never or occasionally prepare previous activities or carry out a follow-up that guarantees subsequent activities are fulfilled, while two out of ten estimates that they do so very frequently or with some frequency.
- The teacher carries out pre-visit activities with some or a great deal of frequency, while post-visit activities are carried out very frequently.

- The frequency with which subsequent activities are carried out is slightly higher than that of previous activities. 59% of the teachers always or very frequently carry out pre-visit activities, while 71.4% of the teachers plan post-visit activities.

A relationship must also be established between the demand for and supply of resources by teachers and the museum.

- According to educators, teachers frequently demand resources, which the museum offers very frequently. A total of 81.3% of educators believe that teachers demand resources frequently or sporadically. A total of 56.3% of educators believe that they always or very often offer resources to schools and 43.7% of teachers believe that they frequently or occasionally offer resources to schools.
- The teachers consider that they demand resources from the museum and the museum offers resources only sporadically/rarely. A total of 58.4% of teachers consider that they never or occasionally demand resources from the museum. A total of 52.7% of the teachers are of the opinion that the museum never or sporadically offers resources and 47.3% quite frequently, very frequently or always.
- The frequency with which resources and materials are offered and demanded at the Alicante Museum (AM) is higher than at the Murcia Museum (MM).
- When the visit is organized through the museum's booking system, the frequency with which the museum offers resources to the school increases. This frequency is even higher at the AM.

In order to answer the research question ('What activities and resources do teachers use in the museum and school within the design of the school visit?') regarding the use of activities and resources, the main conclusion is that the activities that are used most by teachers and educators in the museum (experimentation and artistic workshops, audio-visual observation and viewing tasks and debate or sharing) and by teachers in the classroom (audio-visual viewing) do not guarantee research activities, analysis or reflection activities and, in a certain sense, are coherent with the conclusions already drawn. In other words, the school visit to the museum is a complementary activity, unrelated in most cases to the contents worked on in the classroom and aims to motivate students.

Similar to activities, resources are used by more than half of the teachers who visit the archaeological museum. This may be due to the passive role assumed by the teacher in the museum, which makes him or her unaware of the usefulness of the resources during the course of the visit.

The most commonly used resources, both in the classroom and in the museum, are passive ICT resources which do not involve student participation, followed by worksheets (in the classroom) and objects and material remains (in the museum).

One of the challenges to be addressed is the integration of ICT resources that allow students to participate and interact with heritage elements. Examples could include the use of videogames, work with virtual reality or augmented reality; web applications, such as working with 'Kahoot!' or Plikerts; cooperative work or problem-solving using QR codes; work with virtual museums, etc.

As [48] maintains, in order to promote the use of the socialising perspective of the postmodern museum's heritage, proposals should be generated for interaction between heritage and heritage elements "physically or virtually, through workshops, dramatizations and the use of social networks and their associated tools" (p. 88).

Santacana et al. [36] indicate that what is most relevant for a museum to adapt to a teenage public is its predisposition to interact with young people, transforming itself from a passive exhibition to an active laboratory. To this end, the authors consider it necessary to encourage the use of resources that are of interest to young audiences, including "analogue experimentation resources" and digital resources, demonstrations of experiments, game tasks and activities that complement manipulation. Digital resources include digital games, tablets, virtual reality and augmented reality, as well as experimentation room resources.

## 5. Study Limitations

Having presented the conclusions of the study, it is necessary to establish its limitations, which can be understood as opportunities for improvement required by the research:

- To analyze the educational practices carried out in these non-formal educational contexts.
- To triangulate the results with mixed designs incorporating the point of view of the educational agents, by way of qualitative techniques such as in-depth interviews, group discussions and participant observation.
- To verify whether the post-visit activities which the teachers claim to carry out are indeed put into practice and what their teaching methodology consists of (contents, assessment indicators, standards, competences, methodology, assessment, organization of spaces, times, etc.).
- To understand the perspective of the pupils regarding the conception they hold of museums, school visits to museums, and the development of the activities around the field trip to the archaeological museum.
- To compare the teachers' and educators' perspectives with those of the pupils.
- To enrich the results of this research with other advanced multivariate analyses, such as regression analysis and confirmatory factor analysis, among others.

From the conclusions of the study, it has also been possible to detect the possible needs required of the educational community to improve collaboration and the educational use of school visits to the archaeological museums:

- Better training of museum educators and teachers so that they are able to understand their legal and curricular responsibilities, are aware of the teaching potentials of a museum visit (purposes, use of resources and activities, etc.) and learn how to use the visit to the archaeological museum as a field trip to work on history.
- To design strategies that facilitate collaboration between the museum and the school in preparation for the school visit to the museum or to use this space as a learning resource in the classroom.
- To encourage the utilities and tools necessary to use museum visits as an experimental outing.
- To establish beforehand, a plan of what should be learned from the visit, how that learning is to be achieved and what tasks each of the agents involved in the visit are to carry out.
- To facilitate the planning of the school visit to the museum by the educational community.
- To guarantee that the activities before and after the school visit are carried out and that the archaeological museum somehow participates in the monitoring of these activities.
- To integrate the school visit to the archaeological museum as a learning content or resource within a didactic or teaching unit or within a learning project (learning based on problems, service learning, etc.).
- To encourage the use of ICT resources that allow students to participate and interact with heritage elements.
- To draw up an intervention program that favors and provides resources and support for promoting strategies aimed at the design of collaborative learning projects between museums and schools.

In short, it is necessary to design strategies that enable collaboration between museums and schools to be promoted for the preparation of the school visit to the museum or to use this space as a learning resource in the classroom: In this way, museums will not only be considered as containers of works or providers of resources and activities, but also as a means for the configuration of collaborative learning projects.

**Author Contributions:** Conceptualization, A.E.-M.; methodology, A.E.-M. and F.-J.S.-P.; software, F.-J.S.-P.; validation, A.E.-M., F.-J.S.-P. and P.M.-M.; formal analysis, A.E.-M. and F.-J.S.-P.; investigation, A.E.-M.; resources, A.E.-M.; data curation, A.E.-M. and F.-J.S.-P.; writing—original draft preparation, A.E.-M.; writing—review and editing, P.M.-M.; visualization, A.E.-M.; supervision, A.E.-M.; project

administration, P.M.-M.; funding acquisition, P.M.-M. All authors have read and agreed to the published version of the manuscript.

**Funding:** This research was funded by Ministry of Science and Innovation, the Spanish Research Agency and the European Fund for Regional Development of the EU, Grant Number PGC2018-094491-B-C33 and MCI/AEI/FEDER, UE and by Seneca Foundation—Science and Technology Agency of The Region of Murcia, Grant Number 20874/Pi/18.

**Institutional Review Board Statement:** The study was conducted according to the guidelines of the Declaration of Helsinki and approved by the Ethics Committee of University of Murcia (protocol code 1774/2018, date of approval 5 February 2018).

**Informed Consent Statement:** Informed consent was obtained from all subjects involved in the study.

**Acknowledgments:** The museums participating in the research are thanked for giving us their space, time and material and human resources to contribute to the development of this research.

**Conflicts of Interest:** The authors declare no conflict of interest.

**Appendix A. The Questionnaire MUSELA DO**

24. In the design of the school trip to the archaeological museum, have you prepared any activity?

No ☐ Yes ☐ If so, indicate the activities, the space and time where it was carried out.

| Type of Activity | Museum | Classroom | Before | During | After |
|---|---|---|---|---|---|
| Experimentation or artistic workshop | ☐ | ☐ | ☐ | ☐ | ☐ |
| Role Play or theatrical representation | ☐ | ☐ | ☐ | ☐ | ☐ |
| Problem-solving cooperative tastks | ☐ | ☐ | ☐ | ☐ | ☐ |
| Treasure hunts | ☐ | ☐ | ☐ | ☐ | ☐ |
| Debate, brainstormings, round table discussions | ☐ | ☐ | ☐ | ☐ | ☐ |
| Observation tasks and/or handling of objects | ☐ | ☐ | ☐ | ☐ | ☐ |
| Fieldwork | ☐ | ☐ | ☐ | ☐ | ☐ |
| Data representation | ☐ | ☐ | ☐ | ☐ | ☐ |
| Interviews of experts or family members | ☐ | ☐ | ☐ | ☐ | ☐ |
| Viewing of audio-visual materials | ☐ | ☐ | ☐ | ☐ | ☐ |
| Creation of a classroom museums or exhibition of objects | ☐ | ☐ | ☐ | ☐ | ☐ |
| Others: ___________________________. | ☐ | ☐ | ☐ | ☐ | ☐ |

25. As a complement to the fieldtrip, have you used or are you going to use any material or resource?

No ☐ Yes ☐ If so, indicate the resource, the space and time where it was carried out.

| Type of Resources | Museum | Classroom | Before | During | After |
|---|---|---|---|---|---|
| Web pages, audiovisual videos, photographs | ☐ | ☐ | ☐ | ☐ | ☐ |
| Video games, apps, QR codes, augmented reality, virtual museum | ☐ | ☐ | ☐ | ☐ | ☐ |
| Books, interviews, people from the immediate environment | ☐ | ☐ | ☐ | ☐ | ☐ |
| Worksheets | ☐ | ☐ | ☐ | ☐ | ☐ |
| Objects and material remains | ☐ | ☐ | ☐ | ☐ | ☐ |
| Models or reproductions | ☐ | ☐ | ☐ | ☐ | ☐ |
| Cartographic representations, maps | ☐ | ☐ | ☐ | ☐ | ☐ |
| Other:___________________________. | ☐ | ☐ | ☐ | ☐ | ☐ |

## Appendix B. The Questionnaire MUSELA EDU

20. During school visits, are there any activities?

No ☐ Yes ☐ If so, indicate how often you use them.

| Type of Activity | 1. Never | 2. Punctually/Rarely | 3. With Some Frequency | 4. Very Often | 5. Always |
|---|---|---|---|---|---|
| Experimentation or artistic workshop | ☐ | ☐ | ☐ | ☐ | ☐ |
| Role Play or theatrical representation | ☐ | ☐ | ☐ | ☐ | ☐ |
| Problem-solving cooperative tastks | ☐ | ☐ | ☐ | ☐ | ☐ |
| Treasure hunts | ☐ | ☐ | ☐ | ☐ | ☐ |
| Debate, brainstormings, round table discussions | ☐ | ☐ | ☐ | ☐ | ☐ |
| Observation tasks and/or handling of objects | ☐ | ☐ | ☐ | ☐ | ☐ |
| Fieldwork | ☐ | ☐ | ☐ | ☐ | ☐ |
| Interviews of experts or family members | ☐ | ☐ | ☐ | ☐ | ☐ |
| Viewing of audio-visual materials | ☐ | ☐ | ☐ | ☐ | ☐ |
| Others: _________________________. | ☐ | ☐ | ☐ | ☐ | ☐ |

21. As a complement to the visit, do you use any material or resource?

No ☐ Yes ☐ If so, indicate how often you use them.

| Type of Resources | 1. Never | 2. Punctually/Rarely | 3. With Some Frequency | 4. Very Often | 5. Always |
|---|---|---|---|---|---|
| Web pages, audiovisual videos, photographs | ☐ | ☐ | ☐ | ☐ | ☐ |
| Video games, apps, QR codes, augmented reality, virtual museum | ☐ | ☐ | ☐ | ☐ | ☐ |
| Books, interviews, people from the immediate environment ... | ☐ | ☐ | ☐ | ☐ | ☐ |
| Worksheets | ☐ | ☐ | ☐ | ☐ | ☐ |
| Objects and material remains | ☐ | ☐ | ☐ | ☐ | ☐ |
| Models or reproductions | ☐ | ☐ | ☐ | ☐ | ☐ |
| Cartographic representations, maps | ☐ | ☐ | ☐ | ☐ | ☐ |
| Others: _________________________. | ☐ | ☐ | ☐ | ☐ | ☐ |

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
