# Peer review of "The Use of Activities and Resources in Archaeological Museums for the Teaching of History in Formal Education"

_sustainability, doi:10.3390/su13084095_

Round 1
Reviewer 1 Report
The manuscript develops the description of interesting research focused on the use of activities and resources for the teaching of history based on archaeological museums. It is a valuable contribution to didactics of social sciences and humanities. However, we believe that several changes and corrections are necessary for its publication in a journal such as Sustainability:
- Precisely this is one item that authors should develop more in their work, to establish a clearer connection between the general objectives of the journal and the manuscript. We assume it has been sent to a special issue related to sustainable education (we do not have information on which monographic issue it has been submitted to), but this connection is hardly showed. The concept of sustainability is only mentioned at the beginning of the introduction, in paragraphs 24-32 and 33-41, but is not taken up again elsewhere in the document. It is necessary to establish bridges between the journal and this research, to go deeper into this concept, and at least in the discussion and conclusions sections it should be dealt with again.
- Formatting problems should be corrected (font size according to the template, spaces between paragraphs or elements, typos, etc.). For example, this can be seen in lines 59 and 60; in lines 94 (with a colon in a row) and 95; is it necessary to use capital letters in lines 741 to 745; etc. In order not to list these defectors, we recommend a general review by the authors. Also, look at lines 423-424, it would not be necessary to make captures, they could be included directly as text.
- We would also advise to modify all the styles of the tables (especially the shading, which is sometimes confusing) and the colours of the figures, perhaps too bright.
- As for the sections, we start with the "Introduction". Wouldn't it have been better to separate the introductory part with two sections, "Introduction" and "Context"? Precisely in the context section, where the review of the scientific literature would be inserted, much more depth should be given to the sources. There has not been an exhaustive search for previous works, for more research related to the object of study, etc. There are very few references to it. They should include especially from the last two or three years. It requires a more thorough review of the scientific literature.
- In the section on materials and methods, we are especially concerned about the strategies taken to determine the reliability and validity of the research instruments used. Could the authors elaborate on this at sufficient length?
- Regarding the results section, we believe that there should be greater homogeneity in the presentation of the results. We have already discussed the aesthetics and composition of tables and graphs. We should insist on this.
- The discussion section is completely insufficient. There is hardly any contrast with other similar works, from other countries, on what this research contributes, especially concerning others, etc. A much more in-depth work of sources should be carried out to allow for an authentic discussion section.
- The conclusions section should also be revised. Some elements do not correspond to it (it should be a specific and direct concretion of the achievements got with this research) and that would be more appropriate for the discussion section. Besides, typos are still very present (see the reference to Figure 14 showed as Figure 13).
- Regarding section 6, we have serious doubts about its presence in the manuscript. First, because we believe that these are not patents but documents registered in intellectual property offices. Can the authors better specify and confirm this? We do not believe that a separate section is necessary to show this registration; it sufficed to include it when describing the research instruments used.
It is an interesting and valuable manuscript, but it should be improved so that the results got can reach the scientific community.
Reviewer 2 Report
This is an interesting, original and well-structured manuscript, offering some innovative ideas on the use of activities and resources in museums for teaching history in schools.
The use of figures and tables is effective, as they offer a clear picture of the findings. Sections to be improve are methodology and conclusion.
Minor revisions are needed to strengthen the argument.
First, refine and elaborate methodology more. 'Research-based learning project' (p. 2) is too general. Change it to either historical-comparative research methodology or case study.
Second, conclusion (pp. 21-23) is a mixture of discussion and evaluation. Most should be shifted to discussion section.
Revised conclusion should be brief stressing the significance of findings.
Start with 'The study shows...and then list major findings
Reviewer 3 Report
The study submitted here seeks to analyze, comparatively, the types of activities and resources designed and ultimately used by teachers and museum educators in non-formal educational contexts (two archaeological museums). However, it is precisely in this aim that the main weakness of the study lies: what are the specific contributions of the research that justify its contribution to increasing the body of knowledge in the field of sustainability studies in general and, in particular, of international studies on historical education? What is the reason for not analyzing the educational practices carried out in these non-formal educational spaces and not only the resource statements and activity designs of the educational agents?
A new approach to the objectives of this study is strongly recommended, which could be useful as a first phase of research, and a new wording, considering the following aspects:
1. Lines 162-165. "To develop an approach which allows for the appropriate use of activities and resources offered by museums" and "to contribute towards the development of the teaching and learning process of history in formal education" are two distinct research objectives. Although related, in this paper they are presented as two parts of the same unit.
2. If four research questions are formulated, why are two hypotheses also included? Likewise, the formulation of the hypotheses is incorrect, so it is recommended that they be carefully revised in the context of a quantitative study with hypothesis testing. An adequate formulation can be found, however, in lines 449-454 / 566-571. It cannot be understood, therefore, why they appear almost at the end of the results section and not at the beginning of the study.
3. Line 184. It is stated that "The study is focused on a descriptive phase". However, the incorporation of hypothesis testing implies that the study is developed at a relational research level. Consequently, a detailed review of the levels of research is recommended for the writing of this section.
4. A comparative analysis according to educational level is recommended. It seems logical to think that the results cannot be common to the important specificities of each educational stage.
5. Lines 264-266. The two versions of a, in principle, validated questionnaire are used. However, no empirical evidence of its reliability and validity in the specific context of this research is provided.
6. Lines 283-288. Why is the Mann-Whitney U test used to compare post hoc differences identified with the Kuskal-Wallis H test? The latter nonparametric test provides these differences without resorting to tests specifically aimed at comparing two population groups.
7. Lines 544-545. Please clarify which statistical tests were used in this section and include them in the "Method" section. What type of test is the nonparametric Chi-squared ANOVA test for independent K samples? We hope that this is a confusion due to mistranslation and that it refers to the nonparametric Chi-squared test and the one-way ANOVA.
8. Please rework the tables incorporated in this study, as they correspond to those generated in SPSS.
9. Considering that a traditional research problem has been developed in the field of educational studies, it is recommended that the selected bibliographical sources be expanded and reviewed from an international perspective (only 31 studies are included).
10. Include a brief section on the limitations of this diagnostic study.
11. Finally, a thorough revision of the English grammar and wording of this work is recommended, whose presentation, in its current state, is truly difficult for English-speaking readers.
Reviewer 4 Report
I can see the interest in and local relevance of this research topic and I am sure this was a very interesting journey and a great effort. However, I found the approach quite descriptive and the long narrative that generated an unnecessarily long and complicated piece - though can possibly be explained by the fact that the authors are not writing in but translating from their native language.
Importantly, I felt that this article does not read as particularly informed by the most current or appropriate literature (not least international), empirical or theoretical; not to mention the educational 'jargon' used that seems descriptive - we would be talking about things like educational aims, learning (and other?) outcomes, skills; we would need clear and informed understandings/uses of concepts such as interaction, participation, active/passive (especially given that activities unfold in physical, intellectual and emotional individual domains). The history learning element doesn't come through strongly which it should as the learning of history is the main objective of these projects/activities - what is specific about history, what are the subject-specific outcomes and skills that should frame such a learning project? Fundamentally, I don't think that this study is clearly framed.
I also note that there are quite a few books and certainly academic journals on the topic of education (curricular or not) in the context of museums (and museum education) and the authors will find some articles that are informative and relevant to their topic.
Round 2
Reviewer 1 Report
The authors have made the recommended major modifications and additions, however the references still need to be revised to be in line with MPDI standards. Thank you for the opportunity to have reviewed this interesting manuscript.
Author Response
All references have been reviewed, modified and adapted to the MPDI standards. See the "References" section in red in the manuscript.
Reviewer 3 Report
Dear Authors,
Thank you very much for addressing, in detail, my suggestions, recommendations and corrections, whose objective is that your research reaches the highest possible scientific rigor and impact.
As a last suggestion, please pay attention to the following aspect:
Despite providing empirical evidence on the validation of the instrument applied, no data are provided about the Confirmatory Factor Analysis (CFA), not to be confused with the exploratory one, a technique that seems to have been applied in this new description. Since it is stated that "the MUSELA questionnaire was analyzed via pilot and confirmatory studies", in case a real CFA was performed, please provide the indicators returned by this analysis. Otherwise, please point out this limitation at the end of the study (section 5).
Best regards,
Reviewer 3
Author Response
Lines 391-401 have incorporated the results of the confirmatory study, that is, the data on the Confirmatory Factor Analysis. Specifically, the results obtained for the MUSELA DO and MUSELA EDU questionnaires of the Exploratory Factor Analysis and Kaiser-Meyer-Olkin (KMO) test are presented.
Reviewer 4 Report
This is a much improved submission, certainly in terms of the clarity of communication, well done on this. I still think that, overall, this remains quite descriptive and provides limited learning about 'the use of ... in the teaching of history..'. History remains quite hidden in the article not least as it is not factored much into the study or concluding remarks.
Though overall this reads better now and makes a tighter case.
Author Response
Lines 43-47 explain the relationship between study and the teaching of history.
In fact, within the learning of history in school, the museum is positioned as a didactic resource of the first order. The use of museum materials and resources, such as the handling of objects, the use with historical sources, will allow the development of activities that mobilize the development of capacities for learning history. The archeology museum space by itself is already a space that arouses curiosity about learning about the passage of time.